# Environmental DNA illuminates the darkness of mesophotic assemblages of fishes from West Indian Ocean

Emmanuel Corse[1,2☉], Marie Gimenez[1,3☉]*, Estelle Crochelet[4], Anaïs Paulin-Fayolle[4], Florian Campagnari[4], Océane Desbonnes[4], Léo Broudic[4,5], Patrick Durville[6,7], Florence Trentin[5], Gabriel Barathieu[7], Clément Delamare[8], Thomas Gautier[7], Camille Loisil[9], Patrick Plantard[9], Sébastien Quaglietti[8,10], Thierry Mulochau[4,11‡], Natacha Nikolic[3,4,12‡]

1 Université de Mayotte, Mayotte, France, 2 MARBEC, Université Montpellier, CNRS, IFREMER, IRD, Montpellier, France, 3 Centre de Recherche sur la Biodiversité et l'Environnement (CRBE), Université de Toulouse, CNRS, IRD, Toulouse INP, Université Toulouse 3 – Paul Sabatier (UT3), Toulouse, France, 4 Agence de Recherche pour la Biodiversité à La Réunion (ARBRE), Saint-Denis, Réunion, France, 5 Vie Océane, La Réunion, France, 6 GALAXEA, La Réunion, France, 7 Deep Blue Exploration, Mayotte, France, 8 Service de Plongée Scientifique, Mayotte, France, 9 Poisson Lune, La Réunion, France, 10 Parc Naturel Marin de Mayotte, Pamandzi, Mayotte, France, 11 BIORECIF, La Réunion, France, 12 INRAE, AQUA, ECOBIOP, Toulouse, France,

☉ These authors contributed equally to this work.
‡ TM and NN also contributed equally to this work.
* mariegimenez34@yahoo.fr

## Abstract

The advent of environmental DNA (eDNA) represents a pivotal advancement in probing mesophotic communities, offering a non-intrusive avenue for studying marine biodiversity at greater depths. Using this approach, we characterized and compared the mesophotic reef fish assemblages of two West Indian Ocean islands, Mayotte and La Réunion, which are characterized by different geo-morphological contexts. The sequences obtained were assigned taxonomically and grouped into molecular operational taxonomic units to perform richness and beta diversity analyses. The functional diversity of the assemblages was assessed using five traits, enabling each sequence to be assigned to a functional entity corresponding to a specific trait combination. On both islands, the fish assemblages exhibited a comparable level of taxonomic and functional richness, consisting mainly of piscivorous and planktivorous fishes. These assemblages were primarily composed of families such as Serranidae, consistent with expectations for the mesophotic zone. However, beta diversity analyses revealed that the two islands exhibited different taxonomic and functional structures. For example, La Réunion was characterized by a greater importance of the Muraenidae, while Mayotte displayed a higher representation of families strongly associated with coral reefs (e.g., Zanclidae or Malacanthidae). These results suggest that depth-related forcing constrains fish assemblages to some extent, but that differences in structure remains determined by other, more local factors, likely linked to the

**Data availability statement:** All relevant data are within the paper and its Supporting Information files, and are also available via https://entrepot.recherche.data.gouv.fr/dataset.xhtml?persistentId=doi:10.57745/L0AC2X

**Funding:** The MesoMay 3 program was funded by the Office Français de la Biodiversité (OFB) and the Direction de l'Environnement, de l'Aménagement et du Logement (DEAL) de Mayotte. The nautical resources and staff of the Parc Naturel Marin de Mayotte were used and contributed to the success of this program. The MesoRun program was funded by the European Union's Life program, the Office Français de la Biodiversité (OFB) and the Agence Française de Développement (AFD) through the Life4Best program. However, the funders had no role in study design, data collection and analysis, decision to publish, or preparation of the manuscript.

**Competing interests:** The authors have declared that no competing interests exist.

geo-morphological contexts of the islands and their habitats. This study also revealed that eDNA is a promising method for studying difficult-to-observe taxa, such as moray eels or lanternfish, and may also be relevant for monitoring species depth ranges. Overall, results highlighted the "local scale", "functionally integrative" and "temporally integrative" characteristics of eDNA for studying mesophotic reef fish assemblages. However, this study also highlights the limitations of reference DNA databases, pointing to future prospects for fully exploiting the potential of eDNA approaches in the mesophotic zones of the Indian Ocean.

## Introduction

Mesophotic coral ecosystems (MCEs) are found in the vertical marine zone ranging from 30 m to 150 m in tropical and subtropical regions [1]. MCEs are characterized by a more stable environment than shallower coral ecosystems [2,3], suggesting they may be less affected by anthropogenic impacts and might serve as refugia considering the vertical connectivity and species overlap (e.g., fishes possessing substantial swimming abilities) between shallow and mesophotic waters [4,5], although this hypothesis remains subject to debate [6]. These observations have led to the emergence of the "deep reef refugia hypothesis" [7] and a growing interest in studying these deep-sea ecosystems, particularly at a time when reef biodiversity is facing erosion. Despite a certain degree of connectivity [7,8], some studies indicate that these ecosystems harbor distinct communities [9–11], but they remain understudied [12]. Additionally, recent evidence suggests that surface disturbances can also affect MCEs [13,14]. In light of the coral reef crisis and the fact that deep reefs harbor unique but understudied biodiversity, it is crucial to better characterize these ecosystems.

It is now widely acknowledged that fishes play an essential role in maintaining the functioning of coral reef ecosystems [15,16], and represent an important source of food and income for millions of people [17,18]. MCEs form critical fish habitats [9,19], particularly for fishery-targeted species [19]. Their fish assemblages differ significantly from those in surface waters, harboring unique and rich communities [9,20,21]. Notably, a significant shift in fish assemblages has been observed above and below a depth of approximately 60 m [5,22], both taxonomically and functionally [10,23]. Several studies highlight a strong structuring of fish assemblages along a depth gradient, in terms of species richness, abundance, and biomass [24,25], as well as in trophic structuring [26,27]. For instance, trophic guilds strongly respond to depth gradients in many locations [28]. Piscivores, planktivores, and invertivores tend to dominate MCEs, whereas shallow reefs tend to be dominated by herbivores [23,27,29,30]. In addition, some families are primarily associated with shallow habitats (e.g., Acanthuridae and Scaridae), while others, Labridae and Serranidae, are primarily associated with deeper habitats [27]. This transition in fish communities along the depth gradient suggests that deep-water habitats exert a strong environmental filter on coral reef-fish assemblages [31,32].

Ecological communities that occupy similar habitats, yet are separated by significant geographical distances, may exhibit different species pools while often displaying functional convergence. This phenomenon can be explained by the concepts of community assembly and environmental filtering, which suggest that comparable habitats impose similar environmental conditions, leading to trait convergence by selecting species that share adaptive traits enabling their persistence in a particular environment [33–36]. Consequently, functional trait-based approaches allow for greater generality in describing ecological communities and offer many advantages for understanding community assembly compared to traditional methods based only on species identities [37]. The importance of environmental filters in generating functionally similar communities has already been observed at the biogeographic scale for coral reef communities [32,38–40]. However, environmental factors that predominate in controlling diversity may vary with scale, following what is known as the hierarchical filter model. For instance, climate and geomorphic context exert strong effects at coarse spatial scales, whereas habitat variables are generally more important at finer scales [39,41,42]. Despite its crucial importance for ecosystem functioning and stability, the functional component of mesophotic fish communities remains largely unexplored and poorly understood at various scales.

Studying and sampling MCEs poses significant challenges due to the unsuitability of conventional methods, such as air diving or dredging on hard substrates, as well as the logistical and financial resources required to sample and survey them [43]. Recent advancements in diving equipment [44] and underwater robotics [45] have enabled data collection on MCEs. Recognizing these challenges, environmental DNA (eDNA) metabarcoding emerges as a promising method for studying fish assemblages in MCEs [46]. Its non-invasive nature and reduced resource requirements make eDNA an efficient alternative [47]. However, studies on fish eDNA in the mesophotic environment are scarce, and the method remains underexplored [48–50]. Given the difficulties in surveying mesophotic diversity, the application of eDNA to unveil ichthyological community compositions in mesophotic ecosystems appears to be a highly relevant. Nevertheless, understanding the spatio-temporal framework obtained with eDNA is a recent topic in mesophotic areas. While some studies suggest false positives linked to the transport of allochthonous DNA [51], many recent articles provide evidence that the vertical and horizontal dispersal of eDNA between communities is limited in the marine environment [46,52–54], and that eDNA is reliable for detecting vertical and horizontal community shifts [55–57].

MCEs on Indo-Pacific reef slopes present a vast yet enigmatic expanse despite their substantial surface coverage [58–60]. Addressing this gap, scientific programs have been initiated to conduct mesophotic biodiversity inventories on Mayotte and La Réunion islands, identified as biodiversity hotspots in the southwest Indian Ocean [61]. These two islands differed in their geological history and isolation, and some notable differences in taxonomic composition and trophic structuring of assemblages have already been documented, mainly for surface reefs [61]. The present study aims to explore the ichthyofaunal communities of Mayotte and La Réunion islands from both taxonomic and functional perspective, utilizing deep-sea eDNA method. The first objective of this research was to characterize the fish communities inhabiting the MCEs of these two islands, and, secondly, to compare their taxonomic and functional composition and structure. We predicted that the unique geographic context, that shaped environmental factors such as habitat, and evolutionary history of these islands contribute to taxonomic differences, while the depth that characterized MCEs acts as a biogeographic filter. Consequently, fish assemblages are expected to exhibit common features typical of deep reefs according to the environmental filter hypothesis, particularly in terms of functional roles. However, at a finer scale, local context such as the presence of complex habitats, is also expected to influence the structure of fish communities, leading to distinct differences between the islands despite broader biogeographic commonalities.

## Materials and methods

### Study area and sampling method

The study focused on two islands differing in mesoscale (geographical features) and microscale (habitat) contexts. The island of Mayotte, located to the north of the Mozambique Channel (west of Madagascar), is surrounded by a double

barrier reef. The mesophotic zones in this region are predominantly associated with steeply sloping substrates and complex habitats. Mayotte lies at the epicenter of the "Coral Triangle" in the Western Indian Ocean [61] and is influenced by zonal and regional ocean currents that facilitate larval transport and promote intricate inter-reef connectivity [62]. The island's geological antiquity, coupled with the well-developed nature of its reef structures, emerges as a substantial driving force behind the manifestation of extraordinary biodiversity in the region. La Réunion, also situated in the southwestern Indian Ocean but approximately 800 km east of Madagascar, contrasts sharply with Mayotte in terms of geological history and isolation. Unlike the relatively ancient geological formation of Mayotte, La Réunion is characterized by its recent and dynamic volcanic origins [63]. Its mesophotic zone generally features gentle slopes, lava flows, and mud substrates.

A total of ten geographic stations around Mayotte were sampled from November to December 2021 at depths ranging from 68–89 m (Fig 1A), and eight stations around La Réunion were sampled between September 2020 to January 2021 at depths of 80–107 m (Fig 1B). These stations were selected to ensure representative sampling of the known mesophotic habitats of both islands. Niskin bottles are commonly used for deep-water sampling [48] in open environments with weak surface currents. However, water sample collection in mesophotic reefs posed additional challenges, particularly along steep coral reef slopes with strong currents. To optimize sampling efficiency, we developed two custom 8-liter sampling bottles made of polyvinyl chloride to collect large water volumes. This approach was chosen based on evidence that large-volume eDNA sampling with *in situ* filtration is more efficient than smaller-capacity samplers (e.g., 2 liters), as demonstrated by Govindarajan *et al.* [64]. Before each sampling event, the bottles and divers' wetsuits were sterilised by soaking them overnight in tanks containing 10% bleach, then sealed in sterile bags for transport. Divers transported the two 8-L bottles and sealed both end with female's capes at the required depth (S1 Fig). Once collected, the two bottles were brought to the surface and the samples combined into a clean plastic box, which was immediately sealed with a watertight lid. The lid was manually pierced to allow filtration of 16-L of water through a silicone tube connected to a peristaltic pump on board the boat. The water was filtered using a Sterlitech filter capsule with a pore size of 0.8 μm. After filtration, the water was removed from the capsule, which was then filled with a preservation solution [65] and transported in a cooler for storage at -20°C before DNA extraction in a laboratory specialized in handling rare and degraded DNA. Note that one sample was collected per station.

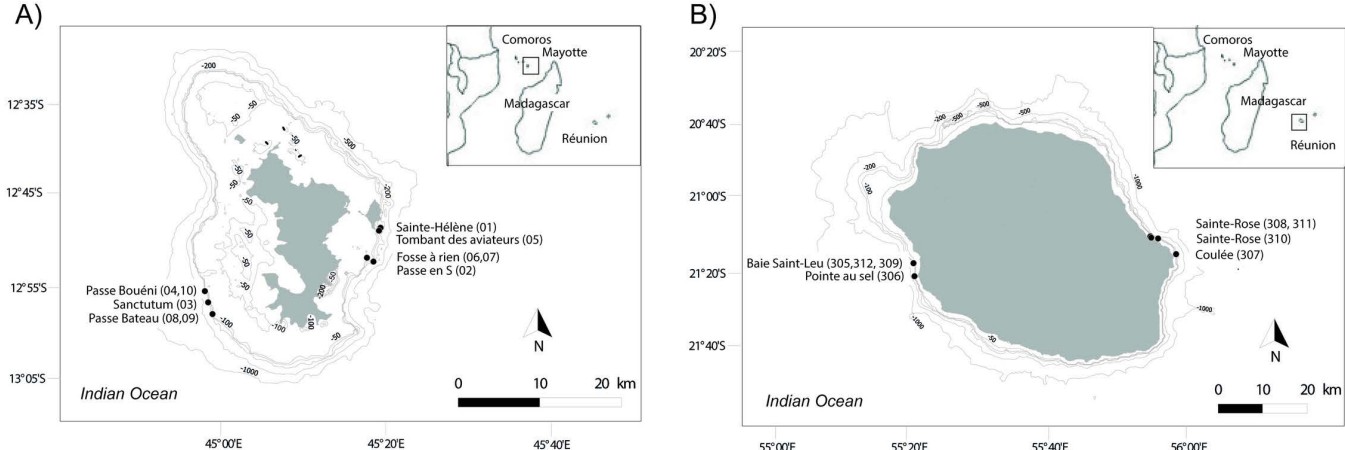

**Fig 1. Maps of environmental DNA samples (black circles). (A)** Depths of 68-89 meters around Mayotte (November to December 2021). **(B)** Depths of 80-107 meters around La Réunion (September 2020 to January 2021). Samples IDs are indicated in brackets. Bathymetric data were sourced from GEBCO and data.gouv.fr. The GEBCO gridded bathymetry data is placed in the public domain and was freely download from https://www.gebco.net/data_and_products/gridded_bathymetry_data/. Source: GEBCO Compilation Group (2024) GEBCO 2024 Grid (https://doi.org/10.5285/1c44ce99-0a0d-5f4f-e063-7086abc0ea0f). Figure created with QGIS Desktop 2.18.0 (https://qgis.org/download/).

To characterize the environment at each station, depth and substrate typology (slope, rugosity, and dominant component) were recorded during dives, and the distance to the coast (in meters – *m*) was calculated *a posteriori* (S2 Table). To compare environmental factors between islands, we performed ordination of environmental data using non-metric multidimensional scaling (NMDS) using *metaMDS* function. The Gower index (using *vegdist* function) was used as the distance metric, suitable for datasets comprising both quantitative and qualitative variables. Subsequently, we assessed correlations between environmental factors and NMDS ordination using the *envfit* function. Variance inflation factors (VIF) were calculated beforehand to avoid multicollinearity [66], ensuring that each variable had a VIF below 5. Mean positional differences between islands in the NMDS were tested using permutational multivariate analysis of variance (PERMANOVA, *adonis* function) [67]. Before testing for differences in environmental composition between islands, dispersion within each island was examined using *betadisper* function, which revealed no significant differences. All analyses were conducted using the vegan R package [68], and all permutations tests with 999 permutations.

## Benchtop to desktop process of environmental DNA

To mitigate contamination risks, eDNA processing was performed at NatureMetrics (Guilford, England) in dedicated eDNA laboratories with unidirectional workflows (from least to most concentrated DNAs). Each step (i.e., eDNA extraction, amplification, library preparation and sequencing) was performed in separate rooms decontaminated with CHEMGENE HLD4L wipes (STARLAB). DNA was extracted using a DNeasy Blood and Tissue Kit (Qiagen), adapted for higher lysate volumes from the filter. Extraction blanks were processed alongside each batch to monitor potential contamination in the extraction process. DNA was purified using a DNeasy PowerClean Pro Cleanup Kit (Qiagen) and quantified with a Qubit dsDNA BR Assay Kit on a Qubit 4.0 fluorometer (Thermo Scientific). Amplifications were performed using the MiFish primer set [69], with 12 replicate PCRs, including a negative and a positive PCR controls (a mock community with a known composition), and by following the steps developed in Díaz *et al.* [70]: (*i*) positive replicate PCR amplicons were purified and pooled for each extraction and prepared into sequencing libraries using a second PCR following methods outlined in Illumina's 16S Metagenomic Sequencing Library Preparation protocol; and (*ii*) the libraries were pooled in equimolar concentrations and sequenced on an Illumina MiSeq with a V3 2 × 300 bp kit with a final concentration of 15 pM with a 20% PhiX control spike. Reads of MiSeq runs were demultiplexed with *bcl2fastq* (Illumina supports) and processed via a custom NatureMetrics eDNA analysis pipeline. Paired-end FASTQ reads were merged with *USEARCH* [71] for each sample. Forward and reverse primers were trimmed from the merged sequences using *cutadapt* [72] and a length filter applied as appropriate for the assay (140–200 bp) to exclude overly long and overly short amplicons from downstream analysis. These sequences were quality filtered with *USEARCH* and dereplicated per sample, retaining singletons to obtain zero-radius OTUs (hereafter ZOTUs) as part of the denoising step. Unique sequences from all samples were denoised in a single analysis with *UNOISE* [73]. Number of sequences that mapped to ZOTUs was calculated and hence a ZOTU-by-sample table was generated with their relative reads proportion in each sample. Finally, low abundance detections were omitted, with filter thresholds set at 0.025% of the read depth of each sample.

## Taxonomic assignment and functional database

Taxonomic assignments were made *via* sequence similarity searches of the ZOTU sequences against the National Center for Biotechnology Information (NCBI) nucleotide (NCBI nt) database (February 2022). Searches were conducted using the BLASTn algorithm [74,75], with hits required to have a minimum e-score of 1e-20 and cover at least 90% of the query sequence. Automatic assignments were made to the lowest possible taxonomic level where consistent matches occurred, with minimum similarity thresholds of 99%, 97% and 95% for species, genus, and higher-level assignments, respectively. Country-based validation and cross-checking with GBIF (Global Biodiversity Information - https://www.gbif.org/) occurrence records were used to manually improve identifications in cases of equally strong reference matches (rgbif) [76]. Special attention was given to key taxa (highly abundant or known for their important functional diversity), specifically

Serranidae, Caranguidae, Holocentridae, Apogonidae, Caesionidae and Muraenidae. For ZOTUs belonging to these families, additional phylogenetic analyses were performed to improve the final taxonomic assignment level of each ZOTUs. ZOTU belonging to these families and their ten best matches from GenBank (excluding matches with query coverage below 90%) were grouped by family in FASTA files. These files were aligned and analyzed in MegaX [77] to achieve phylogenetic inference. The phylogenetical trees were reconstructed using the Neighbor-Joining method with the complete deletion option, and node robustness was assessed throught bootstrap tests (5000 replicates). Taxonomic assignments were deduced from the apical topologies of each tree according to the most common ancestor of the ZOTUs and their closest referent sequences. At the end of the procedure, each ZOTU was assigned to a final taxonomic assignment combining automatic and phylogenetic methods, prioritizing the latter in cases of incongruence.

Based on the functional database of Mouillot *et al.* [78], five qualitative traits linked to key ecological functions were considered: diet (five categories, see details in S5 Datasets), activity patterns (diurnal, nocturnal, or both), schooling behaviour (five ordinal categories ranging from 1 to < 50 individuals), position in the water column (benthic, pelagic, or both, including bathypelagic). Body shape (eel-like, elongated, fusiform/normal, short and/or deep) was also considered rather than fish size. Functional traits were assigned iteratively, starting with ZOTUs identified at the species level using the database of Mouillot et al. [78]. For ZOTUs assigned only to genus or family, traits were imputed by selecting the most prevalent trait values within their respective taxa. ZOTUs assigned to higher taxonomic levels were excluded from functional analyses. In cases of missing values (*NA* entries), supplementary information from FishBase (*fishbase.org*) was used to complete the database, (e.g., *Luzonichthys* or *Diaphus* genus). Each ZOTU was ultimately assigned to a functional entity (FE) representing a specific combination of categorical traits, using the *mFD* package [79].

## Diversity analysis

Unless stated otherwise, all analyses were performed in R studio [80] using the vegan R package [68]. Diversity analyses were based on ZOTU clusters, enabling the assessment of richness within a dataset containing heterogeneous taxonomic assignment levels. Since different ZOTUs may belong to the same species (genetic polymorphism), ZOTUs were aggregated based on their sequence similarity to approximate species-level clustering. To achieve this, clustering analysis of ZOTUs was performed using the *seq_cluster* function (R Bioseq package) [81] with the Complete Linkage Clustering algorithm, which computes the maximal object-to-object distance. The similarity threshold was determined empirically based on a customized MiFish sequence dataset of sequences retrieved from GenBank. The optimal threshold was identified using the *threshOpt* function of the SPIDER package [82], that is useful to assess the barcoding gap by returning the total cumulative errors (false positive and false negative) of identification accuracy for different threshold values (S3 Text for details). Following the clustering analysis, ZOTU occurrences were summed by clusters, generating a new frequency table representing cumulative occurrences of ZOTUs by cluster for each sample. This table was used to assess and compare island diversities. To explore taxonomic and functional diversity, we extrapolated the three orders of Hill numbers, namely richness ($q = 0$, here defined as the cluster richness), Shannon diversity ($q = 1$) and Simpson diversity ($q = 2$), using the iNEXT package [83] with the frequency-incidence data option.

Diversity comparisons between islands were conducted using taxonomic and functional approaches. For each approach, cumulative occurrence tables were calculated for ZOTU by family and by FE, respectively. These tables served to identify taxa and FEs that best characterized the differences between fish assemblages on the two islands. The significance of associations between families or FEs for each island was evaluated using multi-level pattern analysis with the *multipatt* function from indicspecies R package [84]. The tables were then used to generate two sample distance matrices (i.e., Jaccard and the Bray-Curtis dissimilarity indices). Outputs were analyzed using NMDS, and assemblage differences between islands were tested using PERMANOVA (*adonis* function), after confirming conditional homogeneity of dispersion using the *betadisper* function. Families and FEs identified as significantly informative were projected onto multidimensional spaces using the *envfit* function.

Two additional sensitivity analyses were performed on PERMANOVA results: (*i*) excluding the Muraenidae family for both family and FE diversity analyses, given its significance in La Réunion and its singular combination of functional traits; and (*ii*) removing each functional trait one by one from functional analyses (Table 1 and S4 Table). For sensitivity analysis excluding Muraenidae, the results remained unchanged suggesting that this family alone did not drive taxonomic and functional differences between islands (S4 Table). Similarly, removing individual functional traits did not change the results (S4 Table).

## Results

### Habitat factors

The sampled stations on the two islands displayed significantly different habitats (PERMANOVA, $F_{(1,17)}$ = 6.54, $p$ = 0.004). All the environmental factors, except depth, were significantly correlated with the NMDS ordination. Specifically, Mayotte stations were principally characterized by high rugosity and greater distance to the coast (ordination test, $r^2$ = 0.73 and 0.60, $p$ = 0.002 and 0.004, respectively), and secondarily more dominated by coral and a steep slope (ordination test, $r^2$ = 0.44 and 0.58, $p$ = 0.002 and 0.001, respectively). La Réunion was characterized by stations with dominant mud substrate and flatter slope. For both islands, a high slope was associated with less coral or mud cover but with more sessile organisms excluding coral (e.g., sponges) (Fig 2).

### Benchtop-to-desktop workflow of eDNA samples

PCR reactions were successful for all samples, with electrophoresis bands showing strong intensity and aligning with the expected size. Overall, 11–12 successful PCR replicates were obtained for each sample. Both negative and positive controls yielded results within the anticipated parameters. Quality filtering of reads resulted in the loss of <10% of sequences per sample on average across all amplification targets, demonstrating the high-quality nature of the data. Notably, fish sequence reads were conspicuously absent in our extraction and amplification controls, further validating the specificity of our procedures. An ZOTU detected in the sample MesoMay07, attributed to the freshwater genus *Hypophthalmichthys* was removed from the dataset. A total of 503,325 and 970,200 high-quality sequences were retained for the Mayotte and La Réunion datasets, respectively.

**Table 1. Tests of taxonomic and functional differences in fish assemblage structure between islands.** Within-island dispersion and mean island position differences were respectively assessed using a multivariate homogeneity of group dispersions test and PERMANOVA, with the island as the grouping factor. All tests were computed with one degree of freedom and 999 permutations.

| Dataset | Distance matrix | Dispersion tests | | PERMANOVA | | |
|---|---|---|---|---|---|---|
| | | F | *p* | R² | F | *p* |
| All dataset | | | | | | |
| Taxonomic data | Jaccard | 0.57 | 0.47 | 0.13 | 2.48 | **0.004**\*\* |
| | Bray-Curtis | 0.44 | 0.60 | 0.21 | 4.37 | **0.003**\*\* |
| Functional data | Jaccard | 0.25 | 0.63 | 0.15 | 2.85 | **0.001**\*\* |
| | Bray-Curtis | 0.86 | 0.40 | 0.22 | 4.52 | **0.002**\*\* |
| Without Muraenidae | | | | | | |
| Taxonomic data | Jaccard | 1.33 | 0.26 | 0.13 | 2.47 | **0.005**\*\* |
| | Bray-Curtis | 0.63 | 0.53 | 0.20 | 4.00 | **0.003**\*\* |
| Functional data | Jaccard | 0.48 | 0.51 | 0.15 | 2.86 | **0.001**\*\* |
| | Bray-Curtis | 1.54 | 0.25 | 0.20 | 4.09 | **0.001**\*\* |

\*\**p*<0.01.

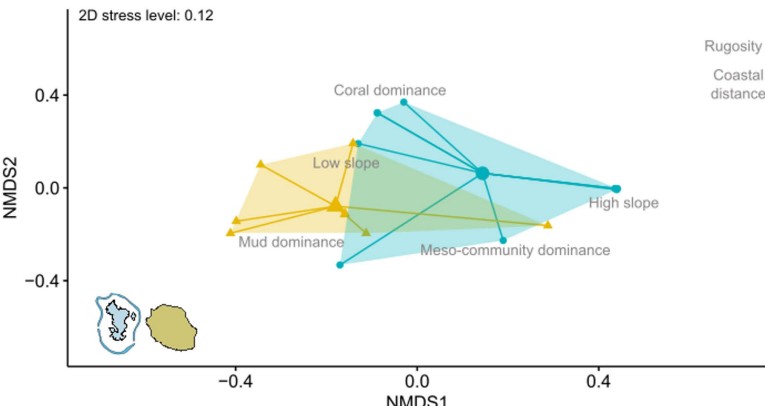

**Fig 2. Comparison of habitat characteristics across islands.** The plot depicts the first two dimensions of the non-metric multidimensional scaling (NMDS) based on Gower distances of environmental habitat variables for each island ($n_{Mayotte} = 10$ and $n_{Réunion} = 8$). Only environmental factors with significant correlation ($p < 0.05$) were fitted and plotted on the NMDS. Coral dominance: Coral was the primary component of the benthic community; Meso-community dominance: Sessile organisms other than coral dominant; Mud dominance: Muddy substrate was the main substrate; High slope: Steep reef slope; Low slope: Flat reef slope. Sample origins are color-coded: Mayotte in blue and La Réunion in beige.

The final ZOTU table was composed of 512 different sequences (301 ZOTUs in Mayotte and 279 in La Réunion, S5 Datasets), with 68 being common between islands. After taxonomic improvement of relevant taxa (specifically Serranidae, Caranguidae, Holocentridae, Apogonidae, Caesionidae and Muraenidae) using phylogenetic analysis (S6 Fig), which concerned 25–37% of ZOTUs for Mayotte and La Réunion datasets, respectively, 94% of the ZOTUs were successfully assigned, at a minimum, to the family level, with 33% achieving identification at the species level (see more details in S5 Datasets and S7 Table). ZOTUs originating from La Réunion exhibited more precise taxonomic assignments compared to those from Mayotte, with 40% versus 28% assigned to species level. This highlights the importance of considering island-specific nuances in the interpretation of the eDNA results, particularly in regions with variable reference database completeness.

### Diversity analysis

To address the significant number of ZOTUs not identified to species, we performed sequence clustering analyses. Based on the 344,035 pairwise distances of the MiFish sequence dataset corresponding to the MiFish amplicon, the optimal distance was determined to be 0.6% (S3 Text) which corresponded to a divergence of ~1 pb. This value was used as the distance threshold for computing clusters of ZOTUs for diversity analysis. The cluster membership of each ZOTU and the MiFish sequence dataset that served to assess the distance threshold are provided in S5 Datasets. Mean estimates of cluster richness ($q = 0$) and diversity indices (Shannon diversity $q = 1$; Simpson diversity $q = 2$) were similar between islands, although slightly higher for La Réunion (e.g., $q = 0$, richness asymptote $= \sim451$ and ~465 for Mayotte and La Réunion, respectively) as suggested by the asymptotic lines in Fig 3.

A total of 61 families were detected. The cumulative occurrence of ZOTU per sample for each family is displayed in Fig 4. Overall, the occurrence of each family among the 18 sampling points was positively correlated with the cumulative occurrence of ZOTUs (Pearson's correlation test, $t = 6.27$, $P_{60} = 4.29e\text{-}08$), showing that the most frequent taxa were associated with a higher number and more frequent ZOTUs. The primary components of the assemblages on both islands were dominated by Serranidae, found in all the 18 samples with a total of 298 cumulative ZOTU occurrences ($16.55 \pm 5.30$ per sample) (Fig 4). A closer examination of this family revealed that Anthiinae constituted the majority of Serranidae ZOTU occurrences ($10.38 \pm 3.25$ per sample for Anthiinae), rather than groupers ($3.38 \pm 1.33$ per sample for

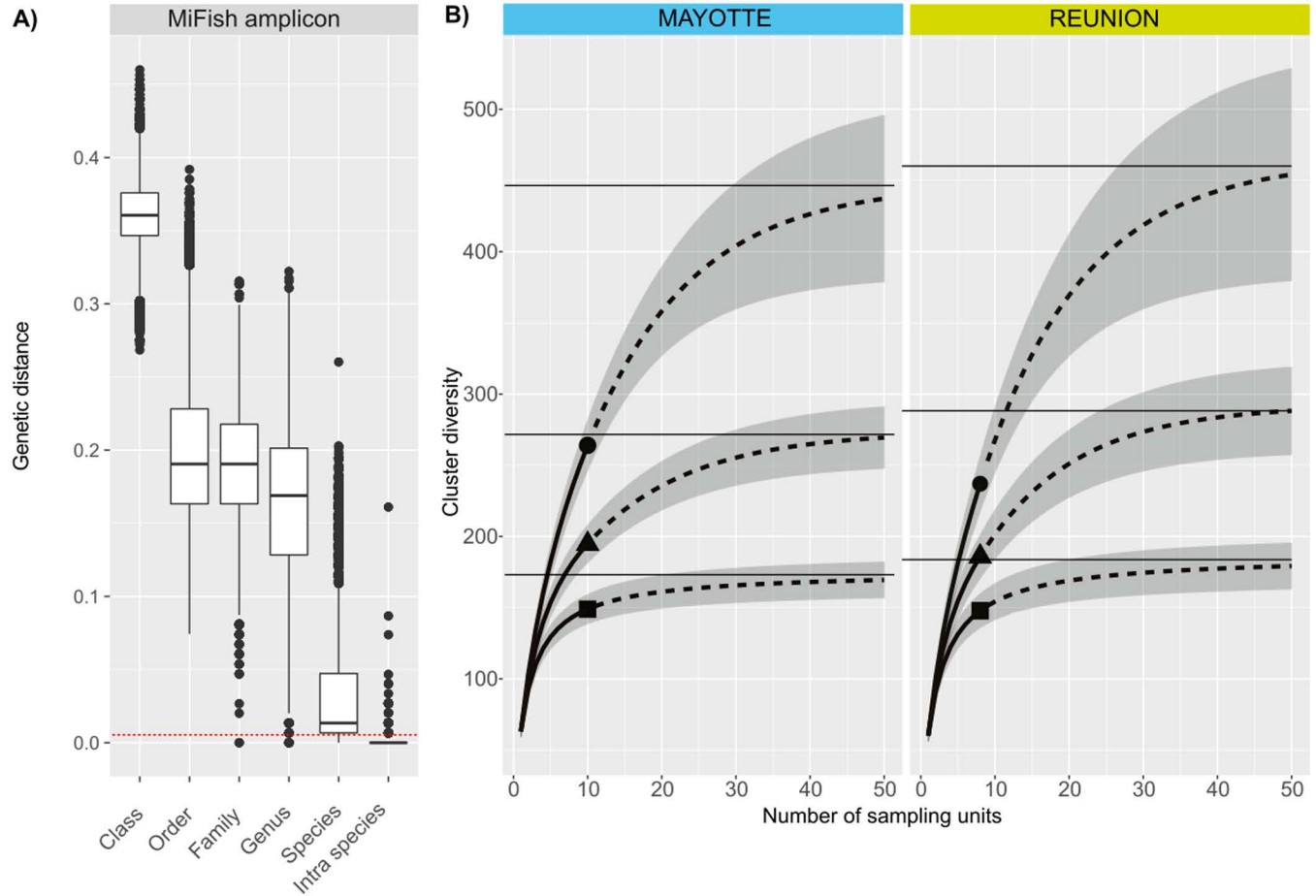

**Fig 3. Diversity analysis.** The boxplot in **A**) represents the distribution of the pairwise genetic distances between sequences of the customized MiFish sequence dataset, categorized by taxonomic comparison type. Class: between sequences from different classes; Order: between sequences within the same class but different orders; Family: between sequences within the same order but different families; Genus: between sequences within the same family but different genus; Species: between sequences within the same genus but different species; Intra-species: between sequences within the same species. The red dotted line corresponds to the divergence threshold defined for clustering ZOTUs. In **B**) curves correspond to the extrapolated rarefaction diversity of ZOTU clusters for each island (incidence-based data) with three orders: richness q = 0 (circle), Shannon diversity q = 1 (triangle) and Simpson diversity q = 2 (square). Horizontal lines represent the asymptotic estimation values for each diversity order.

Epinephilinae). The occurrence among the 18 sampling points and the cumulative ZOTU occurrences of Holocentridae (respectively 16 occurrences on the 18 stations; number of ZOTU occurrences = 84), Acanthuridae (15/18; n = 64), Myctophidae (16/18; n = 61), Lutjanidae (18/18; n = 59), Caranguidae (18/18; n = 59), Muraenidae (16/18; n = 57), Apogonidae (17/18; n = 48), Caesionidae (15/18; n = 43), Balistidae (18/18; n = 43) and Labridae (17/18; n = 34) constituted the common taxa with a mean ZOTU occurrence per sample ranging between 2.38 and 5.25. The remaining assemblages were composed of 50 families that occurred less than 15 times across the 18 samples, with 60% found fewer than five times. Multi-level pattern analysis further revealed that eight families exhibited a significant association with either Mayotte or La Réunion (indicated in Fig 4, see more details in S8 Table 1). Specifically, Mayotte was associated with Labridae (sample occurrence 10/10; cumulative ZOTU occurrence n = 24), Caranguidae (10/10; n = 42), Malacanthidae (7/10; n = 9), Serranidae (10/10; n = 202) and Zanclidae (5/10; n = 5), while La Réunion was associated with Muraenidae (8/8; n = 40), Kyphosidae (5/8; n = 8) and Priacanthidae (4/8; n = 5).

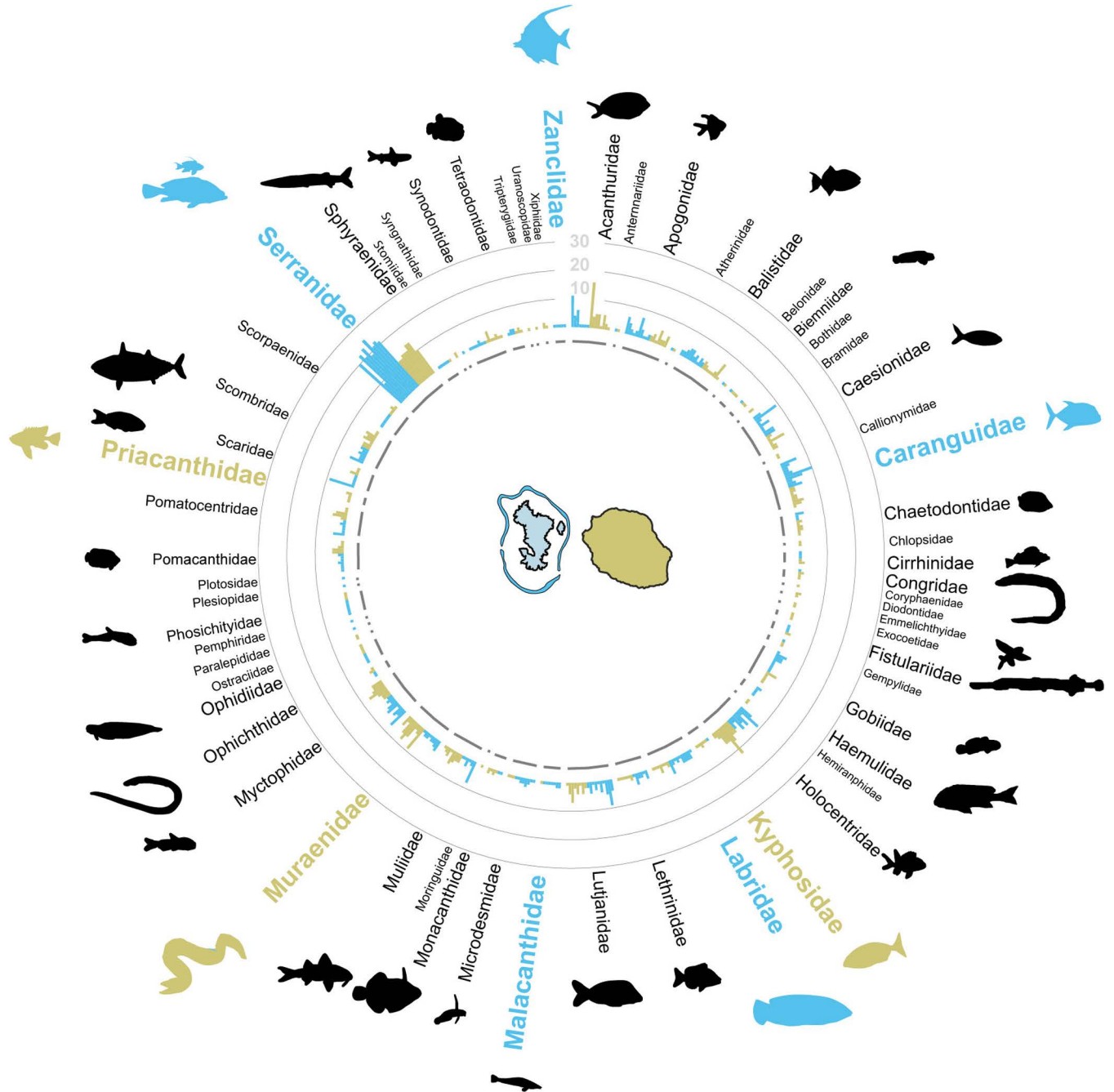

**Fig 4. Taxonomic composition of fish assemblages.** Barplots depict the cumulative occurrence of ZOTUs assigned to each family per sample. The font size of taxa names is proportional to their cumulative occurrence across the 18 samples. Informative taxa significantly associated with one of the islands are indicated by the color code corresponding to the island: Mayotte in blue and La Réunion in beige.

The analysis of functional traits revealed a predominance of diurnal fish and fusiform/normal body shapes on both islands. Diet and position data barplots showed a predominance of piscivores and planktivores on both islands, accounting for more than 60% of occurrences, and more than 75% of occurrences corresponded to taxa associated with benthos

(benthic or bentho-pelagic). Trait values for schooling were more balanced, although solitary individuals or those in small groups (i.e., 3–20 individuals) were more frequently observed (Fig 5). A total of 104 unique combinations of trait values were identified (i.e., 104 FEs, see details in S9 Table), and eight of them were identified as informative for Mayotte and nine for La Réunion using multi-level pattern analysis (Table 2, see more details in S8 Table 2). Informative FEs associated with Mayotte were mainly high-schooling planktivorous or piscivorous taxa (all informative FEs except one, Table 2). In contrast, a more diverse diet, including invertivorous, herbivorous and detritivorous taxa, characterized the informative FEs associated with La Réunion, with the most informative taxa corresponding to fish with eel-like morphology (FE_1 in Table 2).

The taxonomic structure between islands exhibited differences both in terms of quality (presence/absence of family per sample, Jaccard index) and quantity (cumulated number of ZOTU per family and per sample, Bray-Curtis index) with no overlap in both NMDS multidimensional spaces (Fig 6A and 6B). The dispersion of stations across the two islands was not different, but the mean island position differed significantly, with the island effect accounting for 13% and 21% of the overall variation depending on the distance indices (Table 1). A similar pattern was observed for the functional analysis.

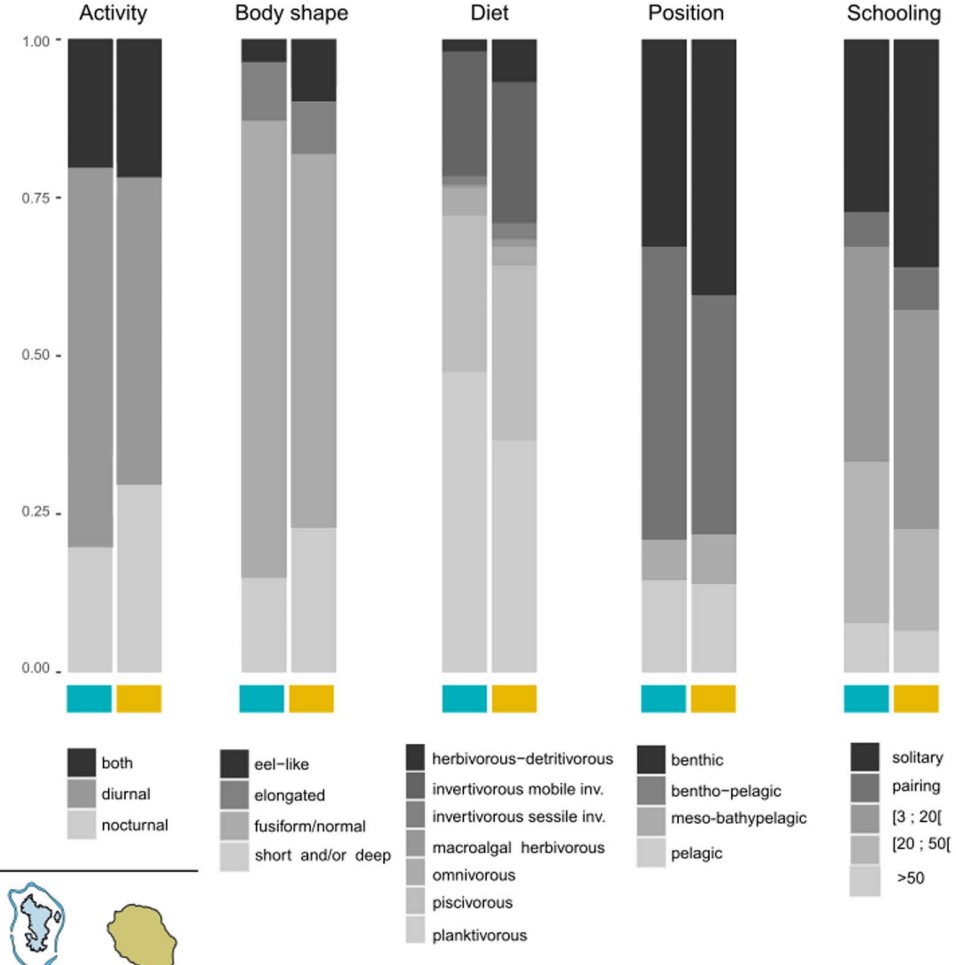

**Fig 5. Functional trait composition of fish assemblages.** Each barplot represents the relative occurrence of trait values by island, based on the ZOTU table for the five functional traits analysed in this study. Barplots for Mayotte are indicated in blue, while those for La Réunion are in beige.

**Table 2. Informative functional entities (FEs).** The table outlines the trait combination values identified as informative (i.e., discriminant between the two islands) through multi-level pattern analysis of cumulative FE occurrence. The "Island" column indicates the specific island to which each FE is associated. Specificity and sensitivity values, calculated using the multipatt function, are provided for each island. FEs are organized according to their island association. Invertivorous mobile inv.: refers to invertivorous fish primarily targeting mobile invertebrates.

| FE ID | Activity | Schooling | Position | Diet | Body shape | Island | Specificity | | Sensitivity | | Test | |
|---|---|---|---|---|---|---|---|---|---|---|---|---|
| | | | | | | | Mayotte | Réunion | Mayotte | Réunion | stat | p |
| FE_39 | nocturnal | [20; 50] | pelagic | piscivorous | elongated | Mayotte | 1 | 0 | 0.5 | 0 | 0.71 | 0.04 |
| FE_47 | diurnal | [20; 50] | bentho-pelagic | planktivorous | elongated | Mayotte | 1 | 0 | 0.7 | 0 | 0.84 | 0 |
| FE_52 | diurnal | solitary | benthic | piscivorous | fusiform/normal | Mayotte | 1 | 0 | 0.6 | 0 | 0.77 | 0.01 |
| FE_72 | diurnal | [20; 50] | benthic | planktivorous | fusiform/normal | Mayotte | 1 | 0 | 0.8 | 0 | 0.89 | 0 |
| FE_38 | diurnal | pairing | bentho-pelagic | invertivorous mobile inv. | elongated | Mayotte | 0.88 | 0.12 | 0.7 | 0.13 | 0.78 | 0.04 |
| FE_10 | both | [3; 20] | pelagic | piscivorous | fusiform/normal | Mayotte | 0.74 | 0.26 | 1 | 0.88 | 0.86 | 0.01 |
| FE_8 | diurnal | [20; 50] | bentho-pelagic | planktivorous | fusiform/normal | Mayotte | 0.72 | 0.28 | 1 | 1 | 0.85 | 0 |
| FE_2 | diurnal | [3; 20] | bentho-pelagic | planktivorous | fusiform/normal | Mayotte | 0.58 | 0.42 | 1 | 1 | 0.76 | 0.03 |
| FE_42 | diurnal | pairing | benthic | herbivorous-detritivorous | short and/or deep | La Réunion | 0 | 1 | 0 | 0.5 | 0.71 | 0.02 |
| FE_43 | nocturnal | [3; 20] | bentho-pelagic | invertivorous mobile inv. | short and/or deep | La Réunion | 0 | 1 | 0 | 0.5 | 0.71 | 0.02 |
| FE_91 | diurnal | [3; 20] | pelagic | invertivorous mobile inv. | fusiform/normal | La Réunion | 0 | 1 | 0 | 0.5 | 0.71 | 0.02 |
| FE_20 | diurnal | pairing | benthic | invertivorous mobile inv. | short and/or deep | La Réunion | 0.09 | 0.91 | 0.1 | 0.75 | 0.83 | 0.01 |
| FE_34 | nocturnal | solitary | bentho-pelagic | invertivorous mobile inv. | short and/or deep | La Réunion | 0.14 | 0.86 | 0.2 | 0.63 | 0.73 | 0.03 |
| FE_64 | diurnal | [3; 20] | pelagic | piscivorous | fusiform/normal | La Réunion | 0.14 | 0.86 | 0.1 | 0.63 | 0.73 | 0.04 |
| FE_73 | diurnal | [20; 50] | bentho-pelagic | macroalgal herbivorous | fusiform/normal | La Réunion | 0.14 | 0.86 | 0.1 | 0.63 | 0.73 | 0.04 |
| FE_24 | both | solitary | benthic | piscivorous | elongated | La Réunion | 0.19 | 0.81 | 0.3 | 0.88 | 0.84 | 0.01 |
| FE_1 | nocturnal | solitary | benthic | piscivorous | eel-like | La Réunion | 0.26 | 0.74 | 0.8 | 1 | 0.86 | 0 |

Note: Specificity refers to the probability that the sampled site belongs to the target island given that the FE has been observed. Sensitivity is the probability of detecting the FE in samples from a given island. Only FEs with p < 0.05 are included.

Dispersion was not significantly different between the two islands, and their mean positions significantly diverged in both NMDS analyses, with no overlap (Fig 6C and 6D; Table 1).

## Discussion

This study demonstrated that our sampling method using eDNA yielded high-quality data with potential for enhanced taxonomic resolution of mesophotic communities, particularly for hard-to-observe families such as Myctophidae or Muraenidae that were not observed in a previous study using a non-baited remote underwater stereo-video system in the area of study [23]. These taxonomic inventories were used to make functional inferences and comparisons of diversity structure. In the following section, we compared the patterns between the two islands to assess how depth constrains in the same way the mesophotic fish assemblage according to the environmental filter hypothesis and the potential role of the antagonist effects of the local context that favors structural differences. We also discussed the methodological implications of our results.

### Common background and divergence in mesophotic assemblages

The patterns observed in the fish communities of both islands can be explained by several successive, scale-dependent filters. At the regional scale, fish communities of the two islands exhibited common characteristics, particularly in terms of functional attributes. Unsurprisingly, we found that they were largely represented by predators, especially piscivores, planktivores, and invertivores, as previously observed in MCEs [28,85]. Trait convergence in reef fish communities driven

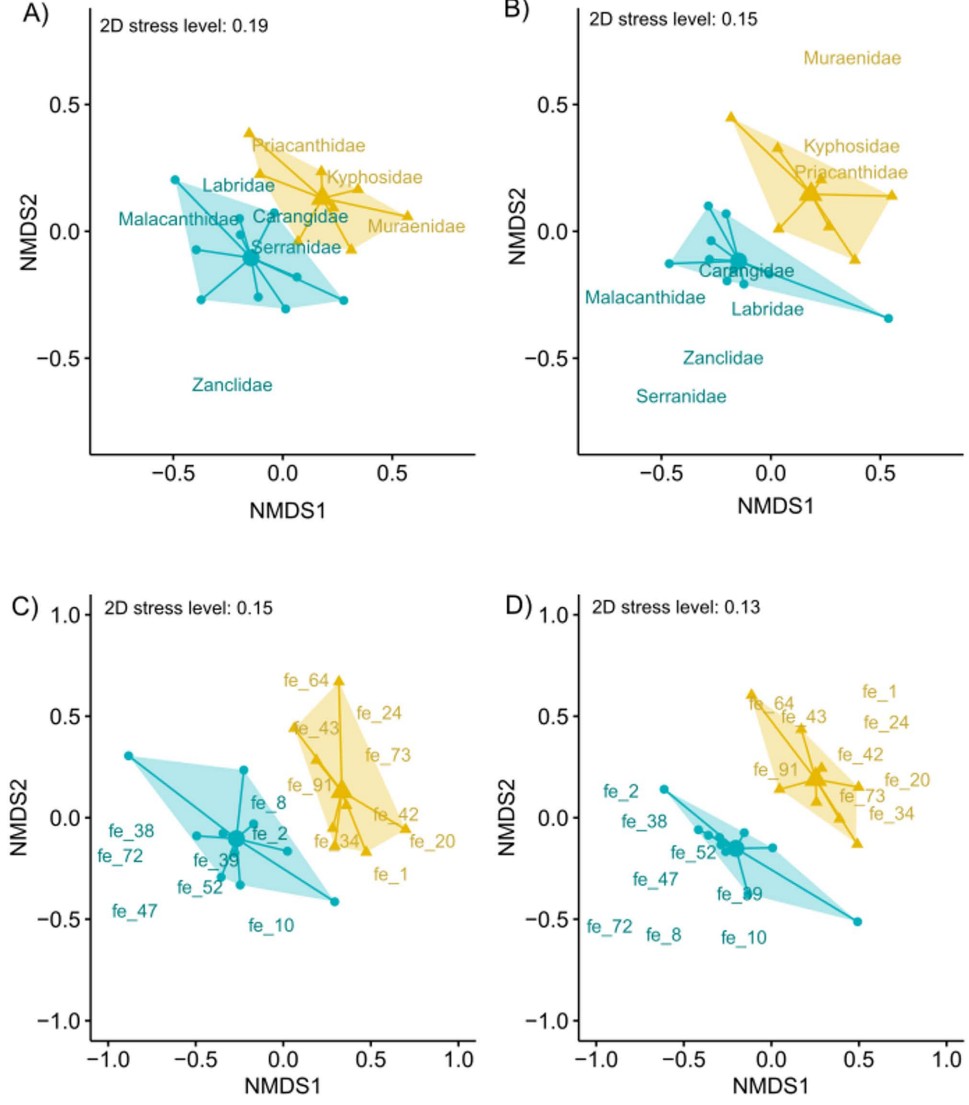

**Fig 6. Taxonomic and functional structure of fish assemblages.** The graph represents the first two axes of non-metric multidimensional scaling (NMDS) of fish taxonomic assemblages using the Jaccard and Bray-Curtis indices in **A)** and **B)**, respectively, and fish functional assemblages using the Jaccard and Bray-Curtis indices in **C)** and **D)**. Sample vectors from island barycenter of each sample and island convex hull are displayed based on their island origin, with Mayotte in blue and La Réunion in beige. Informative taxa and functional entities (FEs) are plotted according to their NMDS scores, with island association indicated by the island color code. The goodness of fit of the two-dimension ordination are indicated for each NMDS by the 2D stress values.

by taxonomically different species has already been observed on a biogeographic scale [23,38,40] and it is particularly true for mesophotic reefs [32], but the communities of the two islands were characterized by different functional entities (FEs). This result suggests that similar trait distributions between islands were not due to identical FEs but rather similar trait proportions [40]. Hence, similar overall trait compositions arise from a variety of FEs, depending on their relative frequencies, particularly when FEs share multiple traits in common, as is the case in deep-reef ecosystems due to environmental constraints that limit the number of viable FEs. These findings align with expectations that depth acts as a strong biogeographic environmental filter, shaping communities by selecting for ecomorphological adaptations in fishes.

From a more taxonomic perspective, the main families occurring were the same between islands, particularly Serranidae, which is not surprising given previous studies [23,27] that identified this family as deep reef specialists. However, this strong prevalence of Serranidae can also be explained by their significant representation by Anthiinae, which form large groups of individuals mostly sedentary. In addition, three of the most represented families are of commercial importance (e.g., Serranidae, Lutjanidae, and Carangidae), as previously observed [19,23,85,86], reinforcing the idea that MCEs are crucial for fishery-targeted species and require specific conservation attention. While we observed some convergence between fish assemblages on a biogeographic scale due to depth, we also observed significant divergence in structure at finer scales. Despite the fact that fish assemblages are mostly composed of the same 11 families (representing two-thirds of ZOTU occurrences), 22 families were found on just one of the two islands, and fish communities of the two islands were characterized by different families. However, the overall richness of communities was comparable, suggesting that the dissimilarity of the assemblage structure can be attributed in part to family turnover [87]. Thus, the amount of species is strongly controlled by depth [32], leading to globally homogeneous communities in terms of richness, but habitat-related factors have an influence, at least locally, not species richness but taxonomic composition within communities.

Geographic location and reef geomorphic features are significant drivers shaping the structure of reef fish assemblages across the Western Indian Ocean [88]. However, part of the observed divergence in the sampled communities can likely also be attributed to habitat differences, as evidenced by the fact that most of the fishes sampled on both islands were closely associated with benthic environments, but habitats in Mayotte were dominated by very rough substrates, making them more complex. This habitat context may explain why certain families associated with the reef environment in Mayotte included species typical of reef fishes (e.g., *Zanclus cornutus*, *Malacanthus brevirostris*) as opposed to those in La Réunion [89], although this does not to explain why Muraenidae were relatively more abundant on La Réunion than in Mayotte. Interestingly a comparable study was conducted on the seamounts La Pérouse, distant to 178 km from La Réunion [46]. Although the two zones have different local contexts, since the Mont La Pérouse is covering by detrital habitats, similar trends of fish eDNA results were observed between these areas, particularly when comparing family richness (56 versus 54) and genus richness (102 versus 114). Myctophidae, Balistidae and Muraenidae were among the richest families in both areas. However, noticeable differences concerned the Labridae family that was the richest in the Monts La Pérouse (11 genus versus 4 in La Réunion), while the Serranidae, Scombridae and Holocentridae are richer in La Réunion (respectively 7, 6 and 5 genus) compared to La Pérouse (less than 5 for the three families). As between La Réunion and Mayotte, we therefore observed a share core structure of the taxonomic fish diversity, and a secondary component that differed even between these closed geographic zones. This reinforces the environmental filter hypothesis at a regional scale that canalizes the fish assemblage in mesophotic zone, which is secondarily shaped by local context.

## eDNA for biodiversity prospects in the mesophotic zone

The coherence of taxonomic and functional patterns of fish communities with expected assemblages from mesophotic zones suggests that our data are minimally influenced by potential allochthonous DNA due to vertical mobility from the euphotic zone and accurately represent the "local" fish assemblage. This indicates that eDNA is suitable for studying mesophotic fish communities, as recent studies have suggested [46,90,91].

We observed a strong alignment with the empirical knowledge of species' vertical distribution (fishbase.org data). The case of Myctophidae, known for their nictemeral vertical migration, is particularly illustrative of eDNA's ability to detect species with vertical migration. The depth ranges at night, as indicated in FishBase for all the five identified myctophids (*Diaphus perspicillatus* [0–240 m], *Diaphus splendidus* [40–225 m], *Diaphus brachycephalus* [0–225 m]*, Bolinichthys longipes* [10–150 m]*, Ceratoscopelus warmingii* [20–200 m]), encompassed the depth range of our samples [68–107 m]. Thus, we are confident that the Myctophidae DNA detected in this study originated from "local" release. As sampling was conducted during the day and no Myctophidae were observed by divers, we infer that the detected DNA was released at least the night before, illustrating the temporally-integrated records of eDNA, as DNA can persist for days in seawater [92],

depending on physicochemical parameters such as seawater temperature [93]. In contrast, several species were detected outside their depth range according to FishBase. For example, *Gymnothorax javanicus* and *Gymnothorax meleagris* were detected even though their previously recorded deepest occurrences were less than 60 m. Again, we are confident that the DNA was locally released, as these two species were observed by divers during the eDNA sampling [94]. Therefore, our study illustrates how eDNA can help define species' depth ranges, a topic of growing interest [46,53], as the depth ranges of marine fishes are usually underestimated [95].

To address the insufficient of taxonomic assignment performance in diversity analysis, we clustered ZOTUs by similarity to approximate species sequence clustering [96]. An appropriate clustering threshold was determined by assessing the barcoding gap (i.e., the separation between intraspecific and interspecific divergences) [97] using a customized sequence dataset retrieved from GenBank corresponding to the MiFish amplicon of taxa found in the study area. The similarity threshold was set at 0.6%, a value that encompassed 96% of the intraspecies dissimilarity values of the customized sequence dataset. This value contrasts with the typically arbitrary values [2% – 3%] used for various mitogenomes minibarcodes but aligns with findings by Jackman *et al.* [98] and Milan *et al.* [99] using different 12S minibarcodes, including the MiFish amplicon [0.5% – 1%]. However, while sequence clustering may address insufficient taxonomic assignment during diversity analysis, the problem persists if the goal is to conduct taxonomic inventories, as this requires more exhaustive reference databases. Notably, rarefaction curve analyses for both islands did not reach asymptotes, indicating that exhaustive taxonomic inventories (at equivalent DNA sequencing depth) would require a sampling effort five times greater.

## Conclusion

The emergence of eDNA represents a significant advancement in exploring mesophotic communities, providing a non-invasive method for studying marine biodiversity at greater depths. Although there are still limitations to these approaches for conducting taxonomical inventories of the fish assemblages in the southwest Indian Ocean (e.g., insufficient barcoding references), this study advocates for a broader application of eDNA for MCEs by demonstrating the local scale, the temporal integration, and the functional integration of this approach. We showed that a reasonable sampling effort in these hard-to-access areas is sufficient to characterize differences in structure between environmental contexts. Further research is needed to define a broader picture of the core structure of fish community of the MCE at regional scale and investigate habitat-fish relationships, which are crucial for local environmental management.

## Supporting information

**S1 Fig. Sample collection with 8-liter sampling bottles developed for this study.**
(DOCX)

**S2 Table. Data to characterize the environment at each sampled station.**
(DOCX)

**S3 Text. Optimum threshold of intra-species divergence delimitation using the MiFish amplicon.** The file listed the bioinformatics flowchart followed to assess similarity distribution according to different taxonomic level of divergence.
(DOCX)

**S4 Table. Sensibility analysis of functional multivariate analysis.**
(DOCX)

**S5 Datasets. Raw data for analyses.** Sheet1 = "ZOTU table" corresponded to the ZOTU table where each line corresponded to a unique ZOTU characterized by an ID (ZOTU shared by the two islands had two IDs, one for each island data) and columns corresponded to sample for which ZOTU reads proportion per sample were indicated. For each ZOTU,

we added information on ZOTU clusters membership, details of taxonomic assignment procedure (automatized assignment; results of additional phylogenetical analysis; final taxonomic assignment and its taxonomic level and taxonomy), details of functional traits acquisition followed by trait values and functional entities membership; sheet 2 = "Rawdata La Réunion"; sheet 3 = "Rawdata Mayotte" corresponded to the DNA sequence and the reads numbers per sample associated to each ZOTU and sheet 4 = "Customized MiFish Sequence dataset" corresponded of the customized MiFish sequence dataset served to assess the genetic distance threshold for the ZOTU clustering. (XLSX)

**S6 Fig. Additional phylogenetical analyses of relevant families.** Four separated analyses were conducted: Fig 1 for Holocentridae; Fig 2 for Muraenidae and Caranguidae; Fig 3 Serranidae and Fig 4 Acanthuridae, Apogonidae and Lutjanidae. The evolutionary history was inferred using the Neighbor-Joining method. The percentage of replicate trees in which the associated taxa clustered together in the bootstrap test (5000 replicates) are shown next to the branches. The evolutionary distances were computed using the number of differences method and are in the units of the number of base differences per sequence. All positions containing gaps and missing data were eliminated (complete deletion option). (PDF)

**S7 Table. Summary of the final taxonomic assignment rank of ZOTUs.** This table provides an overview of the conclusive taxonomic assignment ranks of ZOTUs. The "Global" column does not represent cumulative values since certain ZOTUs are shared between the islands. (DOCX)

**S8 Table. Informative taxa and functional analyses.** This file corresponded to the output of *multipatt* R function using respectively the taxa grouping (at family level) and functional grouping. (DOCX)

**S9 Table. List of combined trait values for each functional entity (FE).** (DOCX)

## Acknowledgments

All staff at the Mayotte Marine Natural Park are thanked for their involvement and participation in the MesoMay program, and in particular Clément Lelabousse for his help in setting up and managing this program. Regarding the MesoRun program, in La Réunion, Daniel Boyer, David Paris, Marc Hénon-Hilaire and Eric Crambes are thanked for the nautical resources and the safety of divers on the surface. The marine ecology laboratory of the University of La Réunion, in particular Sophie Bureau, for access to the laboratory located on the port of St Gilles. The University of La Réunion, IRD and CITEB for supporting this program. Julien Wickel and Ronald Fricke are thanked for their help in the expertise of certain specimens. Daniel Swan and Sarah Chordekar are thanked for their help in the analyses of environmental DNA. We also thank the Genetics Laboratory of CIRAD at the UMR PVBMT for welcoming us when we needed their facilities for our experiments.

## Author contributions

**Conceptualization:** Estelle Crochelet, Natacha Nikolic.

**Formal analysis:** Emmanuel Corse, Marie Gimenez, Anaïs Paulin-Fayolle, Natacha Nikolic.

**Funding acquisition:** Thierry Mulochau.

**Investigation:** Emmanuel Corse, Estelle Crochelet, Florian Campagnari, Gabriel Barathieu, Clément Delamare, Thomas Gautier, Camille Loisil, Patrick Plantard, Sébastien Quaglietti, Thierry Mulochau.

**Methodology:** Emmanuel Corse, Marie Gimenez, Anaïs Paulin-Fayolle, Gabriel Barathieu, Clément Delamare, Thomas Gautier, Camille Loisil, Patrick Plantard, Sébastien Quaglietti, Natacha Nikolic.

**Project administration:** Estelle Crochelet, Thierry Mulochau.

**Resources:** Anaïs Paulin-Fayolle, Florian Campagnari.

**Supervision:** Natacha Nikolic.

**Writing – original draft:** Emmanuel Corse, Marie Gimenez, Natacha Nikolic.

**Writing – review & editing:** Emmanuel Corse, Marie Gimenez, Estelle Crochelet, Florian Campagnari, Océane Desbonnes, Léo Broudic, Patrick Durville, Florence Trentin, Gabriel Barathieu, Clément Delamare, Thomas Gautier, Camille Loisil, Patrick Plantard, Sébastien Quaglietti, Natacha Nikolic.

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
