## [Decision Letter · Decision Letter 0]

1 Dec 2024

PONE-D-24-44756Environmental DNA illuminates the darkness of mesophotic assemblages of fishes from West Indian OceanPLOS ONE

Dear Dr. Gimenez,

Thank you for submitting your manuscript to PLOS ONE. After careful consideration, we feel that it has merit but does not fully meet PLOS ONE’s publication criteria as it currently stands. Therefore, we invite you to submit a revised version of the manuscript that addresses the points raised during the review process. I have tow reviews for your submission. The first reviewer has raised several queries and sought better description of methods and results obtained for clarity. Do take note of the comments offered and suitably revise the manuscript for further consideration and review.

We look forward to receiving your revised manuscript.

Kind regards,

Arga Chandrashekar Anil, Ph. D., D. Agr.,

Academic Editor

PLOS ONE

Journal Requirements:

 The MesoMay 3 program was funded by the Office Français de la Biodiversité (OFB) and the Direction de l'Environnement, de l'Aménagement et du Logement (DEAL) de Mayotte. The nautical resources and staff of the Parc Naturel Marin de Mayotte were used and contributed to the success of this program.

The MesoRun program was funded by the European Union's Life program, the Office Français de la Biodiversité (OFB) and the Agence Française de Développement (AFD) through the Life4Best program.  

Reviewers' comments:

Reviewer's Responses to Questions

**Comments to the Author**

1. Is the manuscript technically sound, and do the data support the conclusions?

Reviewer #1: No

Reviewer #2: Yes

2. Has the statistical analysis been performed appropriately and rigorously? 

Reviewer #1: I Don't Know

Reviewer #2: Yes

3. Have the authors made all data underlying the findings in their manuscript fully available?

Reviewer #1: No

Reviewer #2: Yes

4. Is the manuscript presented in an intelligible fashion and written in standard English?

Reviewer #1: No

Reviewer #2: Yes

5. Review Comments to the Author

Reviewer #1: This manuscript reports on bony fish eDNA metabarcoding in mesophotic coral ecosystems (MCE) surrounding two islands in the Indian Ocean, Mayotte and La Réunion. Mesophotic coral ecosystems (MCE) are deepwater reefs at 30-150 m and are less well characterized by traditional survey methods than near surface reefs due to relative inaccessibility. It is therefore of interest as to look at what can be learned with eDNA.

The researchers obtained water samples at 8 sites around La Réunion (80-107m) in Sept 2020 - Jan 2021 and at 10 sites around Mayotte (68-89 m) in November – December 2021. Water samples were filtered on the vessel and stored at -20C. DNA extraction and fish metabarcode sequencing using MiFish 12S primers were done at NatureMetrics.

As described in the report, the two main technical challenges in applying eDNA to MCEs are 1) sparseness of eDNA in deepwater ecosystems which may necessitate sampling volumes larger than typical 1 L and 2) sparseness of reference sequences for tropical fish, which may require an OTU approach with classification at generic or family level rather than species-level identification. These challenges make this work of considerable interest.

My main comment is that the Methods and Results are insufficiently described particularly in regard to the two questions noted above—eDNA sparseness and incomplete reference libraries. At the same time, I thought the Introduction and Discussion are too long, they should be more focused on the results.

MATERIALS AND METHODS

Line 150. This says “Niskin bottles…were employed”, but Line 153 says “we developed custom 8-liter sampling bottles”. Please be more specific, were different amounts collected at different sites? Were some sites collected with divers and others remotely? This is essential to understanding results. You could include this in S2 Appendix. Also the S1 to S9 Appendix files are labeled with letters rather than numbers.

Line 158. Were the 8 liter samples filtered separately?

I didn’t understand the amplification protocol.

Line 193. Does this mean that PCR inputs were adjusted according to Qubit concentrations? What is volume after extraction and what volume was used for each PCR?

Line 194. I don’t understand what “for each PCR plate” refers to. What happened to multi-liter samples? Were they filtered and extracted separately?

Line 197,198. “i” and “ii” seem to say the same thing.

Line 250. What is basis of “length filter as appropriate for assay”? Does this refer to length of MiFish amplicon? The original MiFish paper lists amplicon length of 163-185 bp.

Line 206. What is a singleton in this context? Is a “zero-radius OTU” referring to a single sequence?

Line 211. What does threshold of 0.025% refer to? Percent of reads for an OTU? Percent of reads for a library?

RESULTS

Line 303. Leave out this sentence.

Line 307. What are “minimal sequences”?

Line 308. Please give actual results, not just state “negligible”.

Line 313. You need to make clearer here and elsewhere whether the OTU tables are before or after the application of 0.3% lumping threshold. I ask partly because the OTU tables in S5 Appendix list single sequences for each OTU. Is Appendix S6/F part of how the clustering was done? I didn’t see a description of tree-building in methods.

Line 314. How many OTUs in each dataset after “taxonomic improvement”? I’m not clear on what procedures “taxonomic improvement” refers to.

Line 319. Leave out this sentence, it is a repeat of prior sentence.

Line 325. If MiFish amplicon intraspecies diversity is defined as less than 0.3% , that is 0.3% x 180bp average amplicon length = 0.54 bp. If this math is correct, then how was there any lumping, i.e., calculation says threshold is below 1 bp difference?

I think depositing FASTQ files is standard for eDNA papers, I don't see that was done.

DISCUSSION

I recommend a more detailed comparison of present results with those in reference 48 as that work was conducted using fish 12S metabarcoding on a seamount in western Indian Ocean (closest land mass is La Reunion) at depths of 0 to 200 m. Are there significant differences or similarities in terms fish species and relative abundance?

48. Muff M, Jaquier M, Marques V, Ballesta L, Deter J, Bockel T, et al. Environmental DNA

711 highlights fish biodiversity in mesophotic ecosystems. Environ DNA. 2023;5[1]:56‑72.

Reviewer #2: This is a well written article and a great study. I have not used eDNA previously, but have been contemplating its use for future work. This study is especially informative and I imagine it will be of interest to others as well.

It would be interesting to learn what the authors think about their sample size. How would the addition or reduction of eDNA samples from each island effect the results? It appears that their sample size was suitable for their comparison and I was curious on how they decided on that number.

Along the same lines, how do the authors expect local abundance to affect occurrence of eDNA? How many samples are needed to confirm a taxonomic differences versus a difference in density. Perhaps this is answered through the cumulative occurrence results and I apologize for not understanding.

6. PLOS authors have the option to publish the peer review history of their article (what does this mean? ). If published, this will include your full peer review and any attached files.

**Do you want your identity to be public for this peer review?** For information about this choice, including consent withdrawal, please see our Privacy Policy .

Reviewer #1: No

Reviewer #2: **Yes: ** David Bryan

---

## [Author Response · Author response to Decision Letter 0]

14 Feb 2025

Reviewer's Responses to Comments

Manuscript Title: Environmental DNA illuminates the darkness of mesophotic assemblages of fishes from West Indian

This document outlines the authors’ responses to the reviewers' comments. All revisions and clarifications made in the manuscript are detailed below.

Comments to the Author

1. Is the manuscript technically sound, and do the data support the conclusions?

Reviewer #1: No

Reviewer #2: Yes

Response: We have thoroughly addressed the reviewer's concerns by revising the Materials and Methods and Results sections. Specifically, we expanded details about the sampling protocol, data analysis, and controls employed to ensure scientific rigor. These clarifications highlight that our methodology aligns with best practices and that the conclusions are appropriately drawn from the presented data.

2. Has the statistical analysis been performed appropriately and rigorously?

Reviewer #1: I Don't Know

Reviewer #2: Yes

Response: We expanded the statistical analysis description in the manuscript. The revised text specifies the methods employed, including PERMANOVA, NMDS, and clustering approaches. Additionally, we have added references to supporting files for methodological transparency.

3. Have the authors made all data underlying the findings in their manuscript fully available?

Reviewer #1: No

Reviewer #2: Yes

Response: The dataset supporting the findings is publicly available at the Dataverse INRAE repository. We have updated the Data Availability section to provide the persistent link and ensure full compliance with the journal’s policy.

The updated text reads (lines 526-528):

The dataset can be accessed via the following link:

https://entrepot.recherche.data.gouv.fr/dataset.xhtml?persistentId=doi:10.57745/L0AC2X.

4. Is the manuscript presented in an intelligible fashion and written in standard English?

Reviewer #1: No

Reviewer #2: Yes

Response: To address linguistic concerns, we subjected the manuscript to professional copyediting to ensure grammatical accuracy and scientific clarity. Redundant or overly complex sentences in the Introduction and Discussion sections were revised for conciseness and focus.

5. Review Comments to the Author

Reviewer #1: This manuscript reports on bony fish eDNA metabarcoding in mesophotic coral ecosystems (MCE) surrounding two islands in the Indian Ocean, Mayotte and La Réunion. Mesophotic coral ecosystems (MCE) are deepwater reefs at 30-150 m and are less well characterized by traditional survey methods than near surface reefs due to relative inaccessibility. It is therefore of interest as to look at what can be learned with eDNA.

The researchers obtained water samples at 8 sites around La Réunion (80-107m) in September 2020 - Jan 2021 and at 10 sites around Mayotte (68-89 m) in November – December 2021. Water samples were filtered on the vessel and stored at -20C°. DNA extraction and fish metabarcode sequencing using MiFish 12S primers were done at NatureMetrics.

As described in the report, the two main technical challenges in applying eDNA to MCEs are 1) sparseness of eDNA in deepwater ecosystems which may necessitate sampling volumes larger than typical 1 L and 2) sparseness of reference sequences for tropical fish, which may require an OTU approach with classification at generic or family level rather than species-level identification. These challenges make this work of considerable interest.

My main comment is that the Methods and Results are insufficiently described particularly in regard to the two questions noted above—eDNA sparseness and incomplete reference libraries. At the same time, I thought the Introduction and Discussion are too long, they should be more focused on the results.

Response:

- Introduction and Discussion Length: We have streamlined both the Introduction and Discussion sections to make them more focused on the results. Specifically, we removed redundant information and reorganized the discussion to emphasize key findings and their relevance.

- Methods and Results Details: In response to the reviewer's request, we expanded the Methods section to provide additional clarity on sampling and processing protocols, particularly addressing challenges related to eDNA sparseness and reference libraries. Key updates include: (i) A detailed description of the custom 8-liter sampling bottles and their efficiency in deepwater ecosystems ; (ii) Clarifications on the ZOTU-based approach and phylogenetic analyses to improve taxonomic resolution.

- eDNA Sparseness and Reference Library Challenges: These challenges were addressed directly in the revised manuscript: (i) We provided additional methodological details for managing low eDNA concentrations in mesophotic zones, including our filtration strategy and the rationale for using larger sampling volumes ; (ii) For incomplete reference libraries, we highlighted our effort to improve taxonomic assignments through phylogenetic analyses and the integration of additional reference sequences.

MATERIALS AND METHODS

Line 150. This says “Niskin bottles…were employed”, but Line 153 says “we developed custom 8-liter sampling bottles”. Please be more specific, were different amounts collected at different sites? Were some sites collected with divers and others remotely? This is essential to understanding results. You could include this in S2 Appendix. Also the S1 to S9 Appendix files are labeled with letters rather than numbers.

Response: We clarified in the Materials and Methods section that custom 8-liter sampling bottles were specifically designed and used for mesophotic reef sampling due to challenges posed by steep slopes and strong currents (lines 147-148). Niskin bottles, commonly used in open water, were mentioned for comparison but were not utilized in this study. In addition, the appendix files have been labeled with numbers.

Line 158. Were the 8 liter samples filtered separately? I didn’t understand the amplification protocol.

Response: The 8-liter samples were not filtered separately. Instead, the two 8-liter water samples collected at each station were combined into a clean plastic container immediately upon retrieval. A total of 16 liters were then filtered using a peristaltic pump connected to a Sterlitech filter capsule with a 0.8 µm pore size. This process ensured an efficient capture of eDNA while optimizing filtration volume. Regarding the amplification protocol, 12 replicate PCRs were performed for each sample using the MiFish 12S primers, following NatureMetrics' standardized workflow. We have clarified these details in the revised manuscript (lines 152-156).

Line 193. Does this mean that PCR inputs were adjusted according to Qubit concentrations? What is volume after extraction and what volume was used for each PCR?

Response: Yes, the PCR inputs were adjusted based on Qubit quantification to ensure consistent DNA input across all reactions. Following extraction, the final elution volume for each sample was 200 µl. From this, 0.9 µl of DNA was used per PCR replicate, resulting in a total of 12 replicates per sample (12 × 0.9 µl = 10.8 µl of the elution used). This step was primarily a quality control measure to standardize the DNA input for amplification. These clarifications have been incorporated into the revised manuscript (lines 191-199).

Line 194. I don’t understand what “for each PCR plate” refers to. What happened to multi-liter samples? Were they filtered and extracted separately?

Response: 16-liter samples were processed through one filtration step, where the entire volume was passed through a 0.8 µm pore-sized filter. The DNA was then extracted from the filter. The term “for each PCR plate” refers to the organization of the PCR setup. Multiple plates were required to accommodate the 12 replicate PCR reactions for each sample, and all plates were prepared identically. To avoid potential confusion, we have removed the mention of “plate” from the text, as it was not essential to the description of the methodology. These adjustments have been made in the revised manuscript (lines 191-199).

Line 197,198. “i” and “ii” seem to say the same thing.

Response: Thank you for pointing this out. We have revised the sentence to eliminate redundancy and ensure clarity. The updated text now provides distinct and non-overlapping information for each point (lines 195-199).

Line 205. What is basis of “length filter as appropriate for assay”? Does this refer to length of MiFish amplicon? The original MiFish paper lists amplicon length of 163-185 bp.

Response: The length filter is an internal cutoff applied for the MiFish assay to exclude sequences that are excessively long or short, which could result from artifacts or sequencing errors. This ensures the retention of high-quality sequences while maintaining the minimum length necessary for reliable downstream analysis. It is important to note that we do not assume the maximum amplicon length is strictly defined by existing reference sequences, as natural variation and sequencing inconsistencies can occur. We have added this justification to the revised text for clarity (lines 202-204).

Line 206. What is a singleton in this context? Is a “zero-radius OTU” referring to a single sequence?

Response: In this context, a singleton refers to a ZOTU that appears only once in the analysis (as per the definition provided at USEARCH manual: http://drive5.com/usearch/manual/singletons.html). Unlike many pipelines, singletons are retained in our analysis to preserve rare sequences. A zero-radius OTU (ZOTU) represents a denoised sequence that is uniquely distinguishable from others, encompassing the full sequence space for any amplicon prior to clustering in a dereplicated state. Thus, ZOTUs do not necessarily correspond to single sequences but instead represent unique sequence variants. To avoid ambiguity, we have replaced all mentions of OTU with ZOTU throughout the manuscript.

Line 211. What does threshold of 0.025% refer to? Percent of reads for an OTU? Percent of reads for a library?

Response: The threshold of 0.025% refers to a percentage of the total read depth for each sample. ZOTUs with read counts below this threshold were excluded from downstream analyses to minimize the influence of potential sequencing errors or contaminants. This clarification has been added to the revised text for transparency (line 209).

RESULTS

Line 303. Leave out this sentence.

Response: The suggested sentence has been removed from the manuscript as requested.

Line 307. What are “minimal sequences”?

Response: Thank you for highlighting this ambiguity. In this context, “minimal sequences” referred to sequences lost during the quality filtering process. To provide clarity, we have revised the text to state: “Quality filtering of reads resulted in the loss of <10% of sequences per sample on average across all amplification targets.” This adjustment ensures a precise description of the dataset processing (lines 307-308).

Line 308. Please give actual results, not just state “negligible”.

Response: The term “negligible” has been removed with specific quantitative results (see sentence above). For example, the quality filtering resulted in a loss of 8.78% of reads for the MiFish 12S primer in La Réunion Island.

Line 313. You need to make clearer here and elsewhere whether the OTU tables are before or after the application of 0.3% lumping threshold. I ask partly because the OTU tables in S5 Appendix list single sequences for each OTU. Is Appendix S6/F part of how the clustering was done? I didn’t see a description of tree-building in methods.

Response: Thank you for pointing out the ambiguity regarding the use of OTU and clustering. In the revised version, we have replaced the term OTU with ZOTU throughout the manuscript to reflect our use of zero-radius operational taxonomic units. ZOTUs represent unique sequences in the dataset and can be considered haplotypes, as they originate from a mitochondrial locus.

The dataset presented in S5 is ZOTU-centered, resulting from the denoising of high-throughput sequencing data. The clustering of ZOTUs was performed exclusively for richness analyses (i.e., rarefaction curves) to account for the high number of ZOTUs not assigned to the species level. Each ZOTU’s cluster membership is indicated in the “Cluster” column, and the clustering method is detailed in the “Diversity analysis” section. To address the reviewer’s concern, we clarified in both the Methods (lines 247-256) and Results (lines 325-333) sections the purpose and methodology of ZOTU clustering.

The phylogenetic analyses, as presented in S6 Figs, were conducted to improve the taxonomic assignments of ZOTUs but were not involved in the clustering process, which was an independent step. These phylogenetic trees, available in S6 Figs, are used to refine taxonomic assignments and are reported in the “Phylogenetical_assignment” column of S5 Datasets. To ensure transparency, we have added a detailed description of the phylogenetic methodology to the main text (lines 224-231).

Line 314. How many OTUs in each dataset after “taxonomic improvement”? I’m not clear on what procedures “taxonomic improvement” refers to.

Response: After taxonomic improvement, the number of ZOTUs in each dataset remained consistent, as the process did not alter the total number of sequences but rather improved their taxonomic assignments. Specifically, 93% of ZOTUs were successfully assigned to at least the family level, and 33% were assigned at the species level. The taxonomic improvement was achieved through phylogenetic analyses, which were performed for ZOTUs belonging to taxa of particular interest (e.g., Serranidae, Carangidae). This approach allowed us to refine the taxonomic resolution for ZOTUs with ambiguous initial assignments.

In the original manuscript, this methodology was summarized to avoid overloading the main text. However, in response to the reviewer’s comments, we have added a detailed explanation of the taxonomic improvement process to the revised Methods section. This includes a description of the phylogenetic analyses and the criteria used for refining taxonomic assignments (e.g., bootstrap support values, lines 224-231). Furthermore, a summary of the final taxonomic assignment performance is available in the supplement file “S7 Table”.

Line 319. Leave out this sentence, it is a repeat of prior sentence.

Response: While we acknowledge the reviewer’s observation, we decided to retain the sentence as it provides complementary information to the preceding one. However, to address concerns about redundancy, we have rephrased and shortened the sentence to ensure it adds value without duplicating content. These adjustments have been incorporated into the revised manuscript (lines 319-322).

Line 325. If MiFish amplicon intraspecies diversity is defined as less than 0.3% , that

---

## [Decision Letter · Decision Letter 1]

31 Mar 2025

Environmental DNA illuminates the darkness of mesophotic assemblages of fishes from West Indian Ocean

PONE-D-24-44756R1

Dear Dr. Gimenez,

We’re pleased to inform you that your manuscript has been judged scientifically suitable for publication and will be formally accepted for publication once it meets all outstanding technical requirements.

Kind regards,

Arga Chandrashekar Anil, Ph. D., D. Agr.,

Academic Editor

PLOS ONE

Additional Editor Comments (optional):

Reviewers' comments:

Reviewer's Responses to Questions

**Comments to the Author**

1. If the authors have adequately addressed your comments raised in a previous round of review and you feel that this manuscript is now acceptable for publication, you may indicate that here to bypass the “Comments to the Author” section, enter your conflict of interest statement in the “Confidential to Editor” section, and submit your "Accept" recommendation.

Reviewer #1: All comments have been addressed

Reviewer #2: All comments have been addressed

2. Is the manuscript technically sound, and do the data support the conclusions?

Reviewer #1: Yes

Reviewer #2: Yes

3. Has the statistical analysis been performed appropriately and rigorously? 

Reviewer #1: Yes

Reviewer #2: Yes

4. Have the authors made all data underlying the findings in their manuscript fully available?

Reviewer #1: Yes

Reviewer #2: Yes

5. Is the manuscript presented in an intelligible fashion and written in standard English?

Reviewer #1: Yes

Reviewer #2: Yes

6. Review Comments to the Author

Reviewer #1: (No Response)

Reviewer #2: Thank-you for the work on the revision and your detailed responses to my questions. I recommend to accept.

7. PLOS authors have the option to publish the peer review history of their article (what does this mean? ). If published, this will include your full peer review and any attached files.

**Do you want your identity to be public for this peer review?** For information about this choice, including consent withdrawal, please see our Privacy Policy .

Reviewer #1: No

Reviewer #2: No

---

## [Editor Report · Acceptance letter]

PONE-D-24-44756R1

PLOS ONE

Dear Dr. Gimenez,

I'm pleased to inform you that your manuscript has been deemed suitable for publication in PLOS ONE. Congratulations! Your manuscript is now being handed over to our production team.

Kind regards,

on behalf of

Professor Arga Chandrashekar Anil

Academic Editor

PLOS ONE